# Turmeric and Cumin Instead of Stock Cubes: An Internet Survey of Spices and Culinary Herbs Used in Poland Compared with Historical Cookbooks and Herbals

**DOI:** 10.3390/plants12030591

**Published:** 2023-01-29

**Authors:** Łukasz Łuczaj, Jarosław Dumanowski, Cecylia Marszałek, Fabio Parasecoli

**Affiliations:** 1Faculty of Biology, Institute of Biology and Biotechnology, University of Rzeszów, ul. Pigonia 1, 36-100 Rzeszów, Poland; 2Faculty of Historical Sciences, Nicolaus Copernicus University, ul. Bojarskiego 1, 87-100 Toruń, Poland; 3MSc Program in Biology, Institute of Biology and Biotechnology, University of Rzeszów, ul. Pigonia 1, 36-100 Rzeszów, Poland; 4Department of Nutrition and Food Studies, New York University, 411, Lafayette Street, 5th Floor, New York, NY 10003, USA

**Keywords:** ethnogastronomy, cuisine, ethnobotany, food globalization, internet ethnography, ingredientization

## Abstract

(Background) Culinary cultures undergo dramatic changes due to globalization; however, the shift in the use of culinary spices and herbs is little documented. We aimed to list seasoning products used in contemporary Polish cuisine and assess whether they have roots in historic recipes. (Methods) Via an online questionnaire distributed via social media, we gathered data from 916 respondents from all over Poland. (Results) Altogether 132 single-ingredient taxa of plant or fungi species were mentioned in the questionnaire. Capsicums, black pepper, turmeric, oregano, cinnamon, marjoram, ginger, basil, thyme, and bay leaf were the 10 most quoted seasonings throughout the year. While local herbs are likely to have been long used in local cooking, the most commonly used spices have been known in Poland since at least 17–18th c. However, a few Asian, mainly Indian spices have become mainstream in the last few years. In particular, these are turmeric, cumin, and fenugreek. The seasonings which decreased the most are various seasoning mixes and stock cubes, unpopular due to the large amount of monosodium glutamate and salt. (Conclusions) Polish cuisine is undergoing a strong process of globalization, and curry spices have become standard cooking ingredients.

## 1. Introduction

Spices are an important component in most human cuisines [1,2]. They have been studied in detail as a source of micronutrients or antioxidants but also as a part of the food-medicine continuum, and even as medicine [3,4,5]. They also play an important role in defining gastronomic traditions both for the communities from which they originate and for outsiders; Elizabeth Rozin famously discussed so-called flavor principles, unique combinations of flavors, including spices and sauces, that constitute a “marker, a defining characteristic of cuisine” [6].

We cannot say that all human cultures use spices, as in many hunter-gatherer and primitive agriculturalist cultures their use is non-existent, and in some, even salt is not added to food [7]. On the other hand, their use can now be found in different parts of the globe [1]. While local aromatic plants served as the first seasonings, with the development of trade, some particularly desirable species became expensive goods transported over large distances [1,2,8]. Spices like pepper and cinnamon already graced the tables of the upper classes in imperial Rome, symbolizing not only wealth but also the political and commercial reach of the Empire. Communities all across the Indian Ocean, from the Red Sea to Southern India to the islands of today’s Indonesia thrived thanks to the trade of spices that grew uniquely in specific locations [9]. During the Middle Ages and the Renaissance, the elites went to great lengths to employ spices in their dinners and banquets as forms of conspicuous consumption [10,11]. Later on, empires were built having among their first goal to ensure access to sources of spices [12]. However, the concept of “spice” is far from univocal; cultures had different perceptions of what constituted a spice, depending on time and place. For instance, sugar was considered an expensive and luxurious spice when it first appeared on Western European tables, to the point that medical properties were attributed to it [13]. Over time, however, as its production increased in territories controlled by Western powers, sugar lost its glamor and became a common, accessible ingredient. As contemporary globalization dynamics facilitate the circulation of ingredients, recipes, and ideas, food connoisseurs around the world continue their quest for new and exciting flavors that also contribute to their cultural and social distinction. At the same time, cuisines all around the world have always taken advantage of the local flora, especially herbs and aromatic plants. As interest in local and artisanal customs increases also among culinary professionals and experts, traditional combinations of herbs, berries, and other botanical elements are increasingly re-evaluated and integrated, also in fine dining cuisine [14,15].

One of the main features of medieval cuisine was the abundant use of Asian spices, which also included sugar. They became the basis of distinction through food and a tool of dietetics, which was cultivated in its ancient form by Arab and Jewish scholars from the Middle East and the Iberian Peninsula [16]. The familiar myth of using Asian spices to preserve meat, and especially to mask the taste of stale produce, is late and ahistorical [17].

Medieval people associated roots with paradise; they were one of the most effective tools of distinction and a means of correcting the humoral properties of food. Above all, however, they determined the attractive taste of the dishes. Stories about using exclusive products to mask the taste of spoiled meat, something much cheaper and more easily available, have all the characteristics of a myth. This simple explanation, despite its rather obvious historicity and repeated demythologization, sometimes penetrates into academic discourse [18].

The popularity of Asian spices was not due to a short-lived and superficial fashion. Fernand Braudel saw it as one of the most important examples of a cycle of *longue durée* (namely “long duration”). The fact that “roots” were known before ancient Roman times has led some researchers to consider the phenomenon in an even longer time frame, and to diminish the role of the Arab-Muslim model. Distinction by exoticism, ascribing medicinal and dietary value to spices, a predilection for their taste and aroma, and culinary aesthetics based on contrast and maximum processing are undoubtedly phenomena known since antiquity, but the range of used roots changed significantly and expanded in the Middle Ages, when the use of e.g., North African silphion (silphium) was discontinued, and cloves and nutmeg, unknown to the Romans, gained great popularity [19].

In the mid-seventeenth century, Asian spices were considered bad taste in France, and the new cuisine was based on local herbs and natural taste. However, in many European countries, including Poland, they have been used for much longer. Their distinctive, spicy taste was recognized in Poland as one of the features of its national cuisine. Spicy Eastern spices began to disappear from Polish culinary recipes only from the end of the 18th century. Paul Tremo, the chef of King Stanisław August, contributed to their reduction, proposing a combination of French and Polish culinary patterns and classicist aesthetics of taste in his recipes [20]. By the 19th century, the assortment of hot spices was already limited.

In English and some other languages, spices are often differentiated from aromatic herbs. In Polish, this distinction does exist but is not so pronounced, and in everyday language, the word “przyprawa” (“spice”) is used for all categories of seasoning. Newerli-Guz [21] analyzed official definitions of spices in the Polish legal system. According to the Polish Norm (Polska Norma) spices are plant ingredients or mixes used to improve the taste of dishes or to give them aroma [22,23,24]. This includes both whole and powdered ingredients [23]. The Polish Classification of Products and Services extends this definition to also include vinegar, sauces, flour, powdered mustard seed, processed spices, their mixes, and salt [25]. The definition of ESA further extends the meaning to seasoning, which must contain plant ingredients but may contain chemical substances to improve the aroma or function of the seasoning (e.g., emulsifiers, preservatives, thickening agents) [26]. The definition of spices/seasoning can be extended even further to include ash [8], sugar [13], monosodium glutamate, or extracts of animal origin [21].

The U.S. Food and Drug Administration (FDA) does not define identity standards for spices, but it supplies a guide of terminology that can be used for product labeling [27]. The FDA characterizes spice as an aromatic plant substance, whether in whole form, broken, or ground, whose main function is to season foods; no essential oil or other flavoring element is extracted from the spice. The CAC (Codex Alimentarius Commission) is aiming at standardizing spice quality and descriptions throughout the globe [28]. Given the large diversity of spices and their definitions, this is a work in progress.

Surprisingly, although thousands of papers have been published on the chemical and medicinal properties of spices, very few papers document the contemporary use of seasoning plants. For example, Łuczaj reviewed wild spices and herbs used traditionally in Poland and Slovakia [29], and Motti [30] performed a similar review for Italy. Khanal et al. reviewed spices traditionally used in Nepal [31] and Wu et al. [32] listed spices used for medicinal hot pots in China. Chironi et al. studied spices and herbs currently used in Sicily, Italy [33]. Spices used in a region of South Africa were also recorded [34]. A small study recorded the most commonly used spices in southern Poland [35]. It must be emphasized that we have not found any study that would look at spice use diachronically by comparing the contemporary use of spices in any country with the historical data on spice use from the same area.

There is no systematic overview of the use of all the spices in Polish culinary history. Information on the history of spice usage in this country can be found in a variety of sources [16,20,36,37,38,39,40,41,42,43,44,45,46,47,48,49,50,51,52,53,54,55,56,57,58,59,60,61,62,63,64,65,66,67,68,69,70,71,72,73,74,75,76,77,78,79,80,81,82,83,84]. Recently many historical cookbooks and recipe collections have been republished, published from manuscripts, or even discovered and published in a series titled “Monumenta Poloniae Culinaria” (e.g., [16,20,37,38,39,40,41,42]). Other important sources on the early use of spices are the earliest herbals and economic dictionaries [43,44,45,46,47,48,49,50,51], a recipe collection of the Teutonic order [52], and accounts of Polish royal courts [53,54]. Archaeobotanical studies tracing the presence of spices in latrines or household remnants provide another source [49,50,51]. The 18th and 19th-century cookbooks and economic dictionaries give us an account of further changes and introductions [52,53,54,55,56,57,58,59,60,61,62,63,64,65,66,67,68,69,70,71,72,73,74,75,76,77,78,79,80].

Łuczaj [29] has made an analysis of spices used by Polish peasants in the 19th and 20th centuries, but we do not know to what extent it correlates with the use of seasoning in pre-Medieval Slavic communities. We can presume that juniper pseudo-berries may have been used, as juniper beer was made in northern Poland and some other peri-Baltic regions [85], and remnants of such beer were found in Denmark in archeological sites from a few thousand years ago [86]. Hops may also be of ancient tradition in this area, as well as caraway and horseradish [87,88]. Some other foreign spices and herbs were later introduced between the Middle Ages and the 18th century [36], and will be discussed in detail further in the article.

In the Baroque period, there was a large interest in exotic, mainly Asian spices [36,89,90]. They were expensive and bought only by the rich, but still very popular. Polish food of that time was spicy (though lacking capsicums). Later, in the 18th century, recipes became more bland (parallel to what was happening in France at that time), and south European herbs used for *bouquet garni*, for example, replaced the Asian spices, which were mainly preserved in recipes for sweets and marinades. This trend continued in the 19th century.

The period of Socialism (1945–1989) after World War II caused a certain homogenization of cuisine, the destruction of manor houses, the emigration of aristocracy, and the simplification of cooking ingredients and recipes. During that time, restaurant tables were usually provided with salt, pepper, vinegar, and a bottle of Maggi™ seasoning (liquid seasoning based on lovage extract popular in many European countries). However, some spices and herbs were used in home cooking, for example, bay leaves (soups, marinades), juniper (sauerkraut, meat), caraway (bread, sauces), marjoram, or allspice. At the end of that period, a mix called “pieprz ziołowy” (“herbal pepper”), containing a mix of spices (e.g., coriander, white mustard, horseradish, etc.) was also quite widely available, although used only by some individuals [91,92].

Recently Polish cuisine has undergone dramatic changes. Chefs, producers, and stakeholders in gastronomy, from writers to event organizers, have been trying to dispel the idea that Polish food is bland, pointing to the wide use of spices in its past and the historical presence of various ethnic communities whose cuisine embraced strong flavors and spices [93,94].

Nowadays, Poland is one of the most ethnically homogenous countries of Europe following severe ethnic segregation of the regions of pre-war Poland after World War II. Poles inhabiting the territories that became a part of the Soviet Union after World War II moved to the new territory of Poland in the West, and many Ukrainians from SE Poland were moved to the Ukrainian Soviet Republic. Most of the Jewish population was either killed or emigrated to Israel. Most Germans in what had become Western Poland left for Germany. These demographic shifts resulted in a country inhabited 95% by Poles [95]. This can be seen as a historically tragic phenomenon but makes the analysis of spice use easier.

Nowadays, wide access to the Internet enables data to be gathered via online forms or email questionnaires [96,97]. Nevertheless, the use of such methods of obtaining information and interviewing is still not very widespread in ethnobotany [98,99,100,101,102]. This can probably be explained by the uncertainty of plant identification or the impossibility of gathering herbarium specimens [103]. However, in the instances when the plants in question are identified or widely known, the Internet can facilitate research [101,102].

The aim of our study was to document the current list of spices and culinary herbs used by contemporary Poles as well as the most recent changes in usage. We also aimed at comparing this list with historical data on spices and herbs used in Poland to establish the status of the currently used spices and herbs and to find out which are new to the Polish culinary tradition. Our hypothesis was that some of the spices which are contemporarily used are completely new to Polish cuisine, and there are no traces of their use in older cookbooks, ethnographic reports, or other culinary sources.

## 2. Results

Altogether, 132 single-ingredient taxa of plant or fungi spices were mentioned in the questionnaire as used at least once a year, 83 of which were listed by at least two respondents. Only 55 kinds of spices/herbs were used by more than 10% of the respondents (Table 1). Also, 31 kinds of other seasonings such as spice mixes, non-botanical ingredients (e.g., honey), or sauces (soy sauce) were mentioned. Twenty of them are used by more than one person (Table 2). The mean number of seasoning types mentioned per respondent was 4.3 (SD = 4.6) for spices used at least once a week, and 3.3 (SD = 4.9) for spices used at least once a year but less than once a week, which leaves us with 7.6 types or species of spices used per respondent in the last year.

Capsicum, black pepper, turmeric, oregano, cinnamon, marjoram, ginger, basil, thyme, and bay leaf were the 10 most mentioned seasonings throughout the year, followed by garlic, allspice, cloves, rosemary, cumin, nutmeg, green cardamom, caraway, coriander, and lovage. Besides single-species spices, the respondents frequently mentioned mixes and sauces used as a seasoning. Here curry mixes and soy sauce were mentioned most frequently (Table 2).

The interviewees generally claimed that they use more kinds of spices than 10 years ago or in their childhood (Compare Table 3 and Table 4). On the list of spices mentioned as only recently used, the most frequently mentioned are turmeric, cumin, capsicum products, coriander, nigella, ginger, green cardamom, curry mixes, fenugreek, and ramsons. All of the first 11 products are common ingredients of south Asian curries and other dishes, with the exception of ramsons (wild garlic), a native wild vegetable (Table 1). Most of the differences between the relative frequency of contemporarily used spices and the spices introduced to the respondents’ diets less than 10 years ago are statistically significant (Table 5).

Some seasoning types have become less popular (Table 4). The most commonly mentioned among these are general-purpose soup seasoning products, often referred to by their brand name (Vegeta™, Kucharek™, Maggi™, and Podravka™). For decades, Maggi and Vegeta have been a common addition to soups, sauces, and meat dishes present on every Polish table and used in most restaurants, but now they are gradually being abandoned. Declining use has also been reported for capsicum products, salt, caraway, black pepper, turmeric, marjoram, “herbal pepper” mixes, curry mixes, thyme, Provence herb mix, etc. Although turmeric and capsicum occur in both lists, their use has more often increased than decreased. On average, respondents mentioned 0.42 types of spices they stopped using over the last 10 years and 2.17 new spices. This well illustrates the increase in the use of seasoning ingredients in cooking.

No significant correlation between the number of species of spices mentioned and the social characteristics of respondents was found (age, town size, gender, number of foreign journeys per year).

## 3. Discussion

Discussing the results of our questionnaire, we should start by stating that our sample was biased toward more literate people and those who are interested in food or plants. Most of our respondents were women with university degree living in large cities. We are aware that the frequency of the use of species in the general population may be lower, and new spices may be less frequently used. On the other hand, most of the spices listed by at least 10% of informants (those listed in Table 6) are, according to our observations, widely available in supermarkets, which suggests they are commonly used. The exceptions are cumin and green cardamom (available in some but not all supermarkets), lemongrass and kaffir leaves (available only in a minority of supermarkets), and sumac, which can be bought almost exclusively from small ethnic or specialist shops in cities.

During the Socialist period as well as in the second half of the 20th century, salt and black pepper were the main ingredients used for “spicing” dishes and were always served alongside each other in restaurants. Sometimes vinegar and lovage-based Maggi seasoning were also served, suggesting that many consumers had no qualms about using artificial seasoning. The presented data show that both sweet and spicy *Capsicum* (paprika/chili etc.) are overtaking black pepper in popularity, or at least equaling it (Table 1). Capsicum products are quite a recent import to popular culinary culture in Poland and were first recorded only in 19th-century Hungarian-style recipes (Table 4) [62,74]. The first author of the paper (born in 1972) remembers trying it for the first time in 1984, around the age of 12, from people who went on holiday to Hungary and brought back both fresh red peppers and paprika powder. This was the first time that the author, brought up in the Polish countryside, saw red peppers! This must have been a common experience for many people traveling from Socialist Poland to Hungary between the 1970s and 1990s. Although spicy capsicums are not used in everyday Polish cuisine, they are ubiquitous in ethnic restaurants in every town and widely used. In the meantime, black pepper has been somewhat forgotten by the media as a boring spice from Socialist times, although it is of course widely used in a variety of dishes.

The answer to the question of which spices are treated as novelties in the diet is best illustrated by the Index of Change (Figure 1). This index enabled calibrating the number of answers about new uses with the number of citations for the contemporary use of spices (compare with Table 1, Table 2 and Table 6). Indeed, the spices with the lowest index, i.e., those never or rarely mentioned as used only recently, are those which were present in Polish cuisine before the 19th century (Table 6), e.g., black pepper, allspice, cinnamon, cloves, mint, caraway, mustard, marjoram, thyme, basil. A few of the spices in this group became popular only in the 19th century, such as capsicums (chili, paprika) and vanilla. Some of the spices with a low IC index are Mediterranean herbs, such as oregano, thyme, basil, and marjoram. Most of them have been present in the Polish culinary heritage. Although their use in recipes has intensified, they were not mentioned as ‘novel’. An interesting case is oregano, which became an extremely popular culinary herb in Poland only after the fall of Socialism, in the 1990s and 2000s, when pizza became widely eaten in Poland. Nowadays, 20 or 30 years later, it is seen as of the main ingredients of everyday dishes, and the fourth most commonly used spice after black pepper, capsicum, and turmeric. On the other hand, the spices with the highest index are those which are not recorded in older culinary sources (Table 6). The highest index was achieved by asafoetida, soy sauce, zatar, garam masala, harissa, ground ivy, kaffir, galangal, Sichuan pepper, sumac, and lemongrass. Most of them are Asian spices or their mixes, apart from the African harissa mix and the native ground ivy.

One of the striking findings of our study is the extreme popularity of what can be grouped as “curry spices”. Out of the list of spices that respondents report as only recently used (Table 3), the first 11 places in the ranking are members of this category. These are spices native to southern Asia and used in the cuisine of India, Thailand, and Indonesia [9,104]. The majority have been known since the Middle Ages or Renaissance and used, if not widely, then at least in aristocratic cuisine or as remedies (Table 6). They later became fashionable in the period of spicy Baroque cuisine. Here we should mention nutmeg, cinnamon, cloves, and fenugreek. Interestingly, mace, recorded in Old Polish cuisine, is hardly known in Poland now. However, some Asian spices, though listed in old herbals, have become fashionable only recently. Here we should list turmeric, green cardamom, cumin, and, to a lesser extent, lemongrass and kaffir leaves. These spices have never played any role in Polish cuisine and have become trendy only recently, as growing numbers of Poles have access to foreign restaurants and to long-distance traveling. It must be admitted that many of the Asian spices persisted in old recipes and as a part of the food-medicine continuum, and the present fashion for ethnic foods and spices only activated their use (chives, cinnamon, nutmeg). The wave of fashion for curry spices was preceded by the popularization of oregano in post-Socialist Poland in the 1990s when pizza became a common food. However, this process has been properly documented.

The popularity of turmeric and, to a lesser extent, cinnamon, cloves, and Mediterranean aromatic herbs may stem from their medicinal properties. Turmeric is now a frequent hero in health columns of magazines and blogs, and its use in Poland can even be classified as a strong fashion. The appearance of such fads for using “superfoods”, often on a global scale, is the result of easy access to information, concerns about health among the population [105], and the search for easy fixes to deal with dietary issues.

A surprising finding of our study is the fact that many respondents listed some wild plants which have never been a part of mainstream cuisine (Table 1). Ground ivy was frequently used in peasant food in some regions [15], but rarely appeared in cookbooks and only recently became popular in wild food dishes (Table 6). Ramsons were almost never used in Polish cuisine, but are now widely known and included in products sold by supermarkets. The increasing importance of some wild foods corresponds to the general worldwide trend of rediscovering local or foraged ingredients [14,15]. On the other hand, some of the most important traditionally collected wild seasonings in Polish cuisines, such as caraway, horseradish, and juniper [15,29], are rarely mentioned, applied mainly in traditional or festive dishes, and not included in fusion cuisine recipes.

This fascination of the Polish nation with exotic foods and spices can be contrasted with Croatia (also a Slavic country), where it is much more difficult to find ethnic foods and foreign spices in ordinary places (the first author’s personal observations). A recently performed survey of herbal liqueurs made in coastal Croatia found that out of over 100 species of ingredients only one, i.e., coffee is not grown in Croatia or does not occur in the wild in this country [106]. This contrasts with the long-standing fascination with liqueurs (*nalewki*) in Poland, which are often made using foreign ingredients such as cloves or cinnamon [107,108]. This can partly be explained by the harsher climate in Poland, which gives Poles a narrower choice of plants to cultivate than in Mediterranean Croatia, but also by a certain culinary conservatism in Croatian cuisine and the relatively low number and quantities of spices and herbs used, especially in Dalmatia.

Chironi et al. [33] looked at the contemporary use of spices by residents of Sicily (Italy). Similarly to our study, they found that a large proportion of respondents now use exotic spices. The authors described this phenomenon as “ethnic contamination”, which we think is derogative and judgmental. Similarly to the Polish population, the respondents frequently used cinnamon, cloves, curry mixes, ginger, nutmeg, or turmeric. There was, however, a large difference in the use of some spices. In Italy, saffron, vanilla, and sesame as well as wasabi and poppy seeds were used more frequently. The Italian respondents also mentioned the use of chia, goji berries, and açaí berries as seasoning. These products are already known in Poland but probably not treated as a seasoning, and that is why they were not mentioned. The term “ethnic contamination” used by the authors of the paper from Sicily [33] is an example of an attitude called “gastronationalism”—seeing a nation as bound to a certain static set of recipes, ingredients, and spices [109]. In Poland, two contrasting currents form the new culinary scene. One is the fascination with fusion cuisine, which can be easily observed in Warsaw, with its dozens of vegan, Vietnamese, Thai, Georgian, Lebanese, and Indian restaurants, among others [110]. The increasing use of Asian spices is another expression of this phenomenon, so visible in the results of our study. This current is strongly correlated with the growing number of vegans and vegetarians who are aware of the health benefits and taste values of new spices brought to Polish cuisine [111]. The growth in the use of spices is not only a Polish phenomenon. India is the world’s largest producer of spices, with a 60% share in output and 36% in the global spices trade [112]. Bagal et al. [112] found that between 2000 and 2017 the export of spices from India nearly tripled (on average 24% growth per annum), signifying a large global demand for Indian spices and spices in general. While the exports of black pepper have increased only a little in the past 16 years, the annual exports of chili, turmeric, and cumin have increased over 10, 12, and 30 times, respectively.

Another current is to retrieve old forgotten recipes, often lost during the uniformization of Polish cuisine during Socialism, and to go back to local wild ingredients, such as ramsons, nettle, or ground ivy. Some of these ingredients have been traditionally used (nettle, ground ivy); others, like ramsons, have not [15].

A few years ago, Żwirska et al. [35] studied spices used by residents of southern Poland. The list of most frequently used spices is similar but not identical. For example, the respondents do not mention turmeric, which suggests that, despite its rapidly increasing popularity, turmeric is a novelty in Polish cuisine.

What is the future of the use of spices in Poland? First of all, we predict a further popularization of the use of exotic spices. The data we obtained come mainly from the educated strata of society and those who are particularly interested in cooking. Their experiments and culinary attitudes will be later followed by the remaining part of the population. This was the case with the popularity of pizza and Italian food 20–25 years ago, and now can be observed for Asian food. One may predict that alongside Asian spices which are popular now (such as turmeric and cumin), in the future we will see the increasing popularity of lemon grass, kaffir, galangal, and zatar mix.

There are definitely a few drivers for the increasing use of spices. One is the already mentioned interest in the health properties of spices, another is the search for new flavors and curiosity, and the last one is the feeling that by using exotic spices and rare ingredients the consumers belong to the “upper” half of the society. This is analogous to the perception of species in Poland in the Baroque period [36]. At that time expensive spices were available only to the rich. Now, when spices are cheaper, the culinary gourmets and people aspiring to be posh or unique may turn to use a large number of spices or rare spices. We may call this “ingredientization”, a new term we want to introduce to gastronomic science. We define ingredientization of food as the search and appreciation of rare food ingredients and attempt to include large numbers of ingredients or previously unknown ingredients in the diet. The process of ingredientization can also be expressed by the search for new and often foraged ingredients [14]. The word ingredientization is nearly absent from Google Search and was previously used only in a slightly different context as “ingredientization” of commodities—previously untapped waste materials used for their potential functional and health benefits [113]. The fascination with long lists of ingredients is very characteristic of East Asian cuisines. Asian markets, e.g., in China and Laos, are rich in species of vegetables, fungi, and meat animals [114,115,116]. This search for a long list of culinary ingredients is now also sprouting in Europe.

## 4. Materials and Methods

An online questionnaire was used. It was constructed as a Google Form in Polish on 14.12.2021 (see additional file in Appendix A) and originally advertised on the first author’s Facebook (Meta) profile, which is followed by around 12 thousand people. It was further shared by 28 other Facebook accounts and a few forums devoted to plants and cooking.

Altogether, 916 questionnaires were filled out from all regions of Poland. The population sample was biased toward more literate, well-educated people interested in plants, herbalism, or cooking. Most respondents were females (74%). Both the median and mean age of respondents was 43, ranging from 18 to 85. All 16 regions of Poland were represented, with the largest number from the following voivodeships: Mazowieckie—22%, Podkarpackie—12%, and Małopolskie—11%. As many as 79% of respondents had a university degree. Exactly 50% of respondents lived in large cities with over 100,000 inhabitants. Sixty-eight percent of respondents stated that they had an average income (68%). Most of them have never lived abroad (83%) and cook food every day (71%).

The spices were easy to identify in nearly all cases, as they have one or two commonly used trade or scientific names. The only problems were with clearly distinguishing various capsicum types, and with distinguishing *Illicium* and *Pimpinella* in the case of anise. That is why both capsicum and anise were treated as one category.

The Chi-squared Test [117] was used (Table 5) to test if the frequency of mention of spices in the questions about newly used or abandoned spices is statistically different from the frequency of spices used at least once a year (last column in Table 1).

Spearman rank correlation coefficient [117] was used to measure the correlation between the main sociological data (age, town size, gender, income, educational level, number of foreign journeys per year) of the informants and the number of spices they use at least once a week, and at least once a year.

An Index of Change (IC), introduced by the authors, ranging from 0 to infinity was applied to illustrate the novelty of a particular spice use:IC *=* (N + 1)/(C + 1)
where: N—number of times the spice was mentioned as used only for 10 or fewer years, C—number of times the spice was mentioned as used at least once a year. The index is close to zero for spices that are commonly used since the respondents’ youth, and close to, or even exceeding, 1 for spices in use since less than 10 years ago.

## 5. Conclusions

Contemporary Poles tend to use growing numbers of spice and herb species in their food. The list of frequently used kinds is a mix of three categories: spices traditionally used since at least the Renaissance or the Baroque period, new (mainly Asian) spices which have recently become fashionable, and spices that were rarely used during Socialist times or were abandoned in the 19th and 20th c. but were present in the culinary culture of the Baroque social elites. The increasing popularity of Indian, Thai, and Middle-Eastern spices should be noted, together with the surge in the use of wild ingredients foraged from the surrounding environment, such as wild garlic or ground ivy.

## Figures and Tables

**Figure 1 plants-12-00591-f001:**
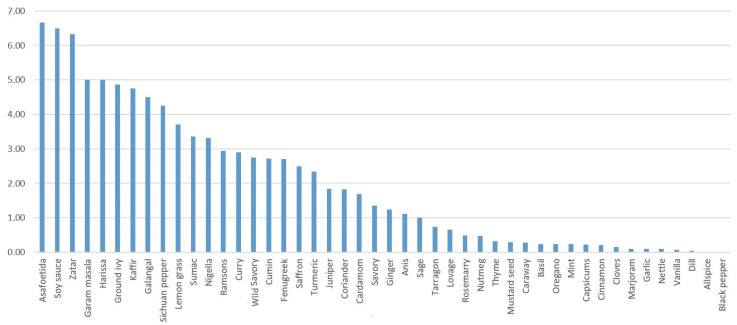
Spices and spice mixes aligned according to the Index of Change (IC), from the most novel in use to those already used 10 years ago.

**Table 1 plants-12-00591-t001:** Types of single-species botanical seasoning mentioned by more than one study participant. Spice mixes were excluded.

English Name	Polish Name	Latin Name	Parts Used	Status	Used More Than Once a Week %	Used at Least Once a Year %	Altogether %
“Capsicums”, i.e., Chilli, Paprika, Cayenne pepper, Gochugaru, and the like	Papryka, chilli, Pieprz cayenne, Gochugaru	*Capsicum* spp.	fruit	i	50.4	8.6	59.1
Black pepper Including White pepper	Pieprz, Czarny pieprz, Pieprz biały	*Piper nigrum* L.	fruit	i	54.8 1.9	1.9 1.6	56.7 3.5
Turmeric	Kurkuma	*Curcuma longa* L.	rhizome (powdered)	i	33.1	17.4	50.4
Oregano	Oregano	*Origanum vulgare* L.	flowering tops	i/c	35.4	13.3	48.7
Cinnamon	Cynamon	*Cinnamomum zeylanicum* J.Presl	bark	i	27.7	19.8	47.5
Marjoram	Majeranek	*Origanum majorana* L.	flowering tops	i/c	32.2	14.8	47.1
Ginger	Imbir	*Zingiber officinale* Roscoe	rhizome	i	29.6	13.2	42.8
Basil	Bazylia	*Ocimum basilicum* L.	leaves	i/c	31.4	10.9	42.4
Thyme	Tymianek	*Thymus vulgaris* L.	flowering tops	i/c	28.2	12.7	40.8
Bay leaf	Liść laurowy	*Laurus nobilis* L.	leaves	i	27.0	9.5	36.5
Garlic	Czosnek	*Allium sativum* L.	bulbs	c	32.9	3.1	35.9
Allspice	Ziele angielskie	*Pimenta dioica* (L.) Merr.	fruits	i	26.2	9.6	35.8
Cloves	Goździki	*Syzygium aromaticum* L.	flower buds	i	9.8	24.8	34.6
Rosemary	Rozmaryn	*Salvia rosmarinus* Spenn.	leaves	i/c	15.3	17.5	32.8
Cumin	Kumin, Kmin rzymski	*Cuminum cyminum* L.	fruits	i	15.1	17.5	32.5
Nutmeg	Gałka muszkatołowa	*Myristica fragrans* Houtt.	seed	i	8.5	21.4	29.9
Green Cardamom	Kardamon zielony	*Elettaria cardamomum* (L.) Maton	fruits	i	10.5	18.1	28.6
Caraway	Kminek	*Carum carvi* L.	fruits	c/w	10.9	15.4	26.3
Coriander	Kolendra	*Coriandrum sativum* L.	fruits and leaves	i/c	12.7	12.7	25.3
Lovage	Lubczyk	*Levisticum officinale* L.	leaves	c	12.1	9.0	21.1
Star anise, anise	Anyż gwiazdkowaty, Anyż	*Illicium verum* Hook.f. & *Pimpinella anisum* L.	fruits	i	2.8	15.7	18.6
Savory	Cząber	*Satureja hortensis* L.	flowering tops	i	6.8	9.6	16.4
Nigella	Czarnuszka	*Nigella sativa* L.	seeds	i/c	7.8	8.6	16.4
Parsley	Pietruszka	*Petroselinum crispum* (Mill.) Fuss	mainly leaves	c	12.2	3.1	15.3
Mint	Mięta	*Mentha* spp.	leaves	c/w	6.0	8.5	14.5
White mustard	Gorczyca (biała)	*Sinapis alba* L.	seeds	c	2.3	11.0	13.3
Dill	Koper, Koper ogrodowy, Koperek	*Anethum graveolens* L.	leaves, fruits	c	7.9	5.3	13.2
Fenugreek	Kozieradka	*Trigonella foenum-graecum* L.	seeds	i	4.1	7.1	11.2
Tarragon	Estragon	*Artemisia dracunculus* L.	leaves	i	1.7	8.7	10.5
Ramsons	Czosnek niedźwiedzi	*Allium ursinum* L.	leaves	w	4.0	6.0	10.0
Sage	Szałwia	*Salvia officinalis* L.	leaves	i/c	2.6	6.4	9.1
Saffron	Szafran	*Crocus sativus* L.	stigmas	i	0.2	7.5	7.8
Vanilla	Wanilia	*Vanilia planifolia* Andrews	fruits	i		6.1	7.4
Nettle	Pokrzywa	*Urtica dioica* L.	young shoots	w	0.8	5.0	5.8
Sumac	Sumak	*Rhus coriaria* L.	fruits	i		4.9	5.7
Lemongrass	Trawa cytrynowa	*Cymbopogon citratus* (DC.) Stapf	leaves	i		4.6	5.1
Kaffir lime	Kaffir	*Citrus hystrix* DC.	leaves	i		3.5	4.1
Wild savory	Czubryca	*Satureja montana* L.	flowering tops	i		2.2	4.0
Ground ivy	Bluszczyk kurdybanek	*Glechoma hederacea* L.	leaves	w	1.3	2.5	3.8
Juniper	Jałowiec	*Juniperus communis* L.	pseudo-fruits	w	2.0	1.1	3.1
Onion	Cebula	*Allium cepa* L.	whole plant	c	2.2	0.8	2.9
Lemon balm	Melisa	*Melissa officinalis* L.	flowering tops	i/c		2.2	2.9
Horseradish	Chrzan	*Armoracia rusticana* G.Gaertn., B.Mey. & Scherb.	roots	w/c	0.8	1.7	2.5
Chives	Szczypiorek	*Allium schoenoprasum* L.	leaves	c		1.2	2.4
Galangal	Galgant, Galangal	*Alpinia galanga* (L.) Willd.	rhizome	i		1.5	2.2
Sichuan pepper	Pieprz syczuański	*Zanthoxylum* spp.	fruit shell	i		1.9	2.2
Sesame	Sezam	*Sesamum indicum* L.	seeds	i		1.5	2.0
Asafoetida	Asafetyda	*Ferula* spp.	roots	i		1.2	1.5
Lavender	Lawenda	*Lavandula* spp.	flowering tops	i/c	0.3	1.1	1.4
Celery	Seler	*Apium graveolens* L.	shoots and roots	c		0.4	1.3
Mushrooms (dried)	Grzyby (suszone)	Fungi	fruiting body	w		1.0	1.3
Ground elder	Podagrycznik	*Aegopodium podagraria* L.	leaves	w		0.8	1.2
Hyssop	Hyzop	*Hyssopus officinalis* L.	flowering tops	i/c		1.0	1.2
Fennel	Koper włoski, Fenkuł	*Foeniculum vulgare* Mill.	whole plant, fruits	i/c		0.4	1.1
Tomatoes (dried)	Pomidory (suszone)	*Solanum lycopersicum* L.	fruits (dry)	i/c		0.5	1.1
Mugwort	Bylica	*Artemisia vulgaris* L.	leaves	w		0.5	0.9
Curry leaf	Liście curry	*Murraya koenigii* (L.) Sprengel	leaves	i	0.0	0.9	0.9
Mace	Kwiat muszkatołowy	*Myristica fragrans* Houtt.	seed coats	i		0.7	0.8
Yarrow	Krwawnik	*Achillea millefolium* L.	leaves	w		0.5	0.7
Long pepper	Pieprz długi	*Piper longum* L.		i		0.5	0.7
Pellitory	Bertram	*Anacyclus pyrethrum* (L.) Link	fruits	i		0.0	0.5
Milk thistle	Ostropest	*Silybum marianum* (L.) Gaertn.	seeds	i/c		0.1	0.5
Wormwood	Piołun	*Artemisia absinthium* L.	leaves	w		0.4	0.5
Wasabi	Wasabi	*Eutrema japonicum* (Miq.) Koidz.	rhizomes	i		0.4	0.5
Olive	Oliwa	*Olea europea* L.	oil	i		0.4	0.5
Black cardamom	Kardamon czarny	*Amomum subulatum* Roxb.	fruits	i		0.4	0.4
Archangel	Arcydzięgiel	*Angelica archangelica* L.	roots	c/w		0.3	0.4
Liquorice	Lukrecja	*Glycyrrhiza glabra* L.	roots	i		0.1	0.3
Rose geranium	Geranium (Pelargonia pachnąca)	*Pelargonium graveolens* L’Hér.	leaves	c		0.3	0.3
Cubeb	Pieprz cubeba	*Piper cubeba* L.f.	fruits	i		0.3	0.3
Perilla, Shiso	Pachnotka	*Perilla frutescens* (L.) Britton	leaves	i/c		0.3	0.3
Avocado	Awokado	*Persea americana* Mill.	powdered seed	i		0.1	0.3
Tamarind	Tamarynd	*Tamarindus indica* L.	fruit pulp	i		0.3	0.3
Hops	Chmiel	*Humulus lupulus* L.	fruits	c/w		0.2	0.3
Ajwain	Ajwain	*Trachyspermum ammi* (L.) Sprague ex Turrill	fruits	i		0.2	0.2
Chervil	Trybula	*Anthriscus cerefolium* (L.) Hoffm.	leaves	c/i		0.1	0.2
Dittany	dyptam	*Dictamnus albus* L.	root	i/c?		0.1	0.2
Cuban oregano	Mięta kubańska, Oregano kubańskie	*Coleus amboinicus* Lour.	green parts	c		0.2	0.2
Tonka beans	Tonka	*Dipteryx odorata* (Aubl.) Willd.	seeds	i		0.2	0.2
Lemon verbena	Werbena cytrynowa	*Aloysia citriodora* Palau	leaves	i/c		0.2	0.2
Barberry	Berberys	*Berberis vulgaris* L.	fruits	i/w?		0.1	0.2
Thai basil	Bazylia tajska	*Ocimum basilicum* L. ‘Horapha’	leaves	i/c?		0.1	0.2

i—imported, c—cultivated, w—wild.

**Table 2 plants-12-00591-t002:** Spice mixes and liquid non-botanical or mixed types of seasoning.

English Name	Polish Name	Used More Than Once a Week %	Used at Least Once a Year %	Altogether %
Curry	Curry	11.6	11.7	23.3
Provence herbs	Zioła prowansalskie	8.0	1.9	9.8
Garam masala	Garam masala	2.0	4.1	6.1
Herbal pepper mix	Pieprz ziołowy	3.5	1.3	4.8
Soy sauce, including tamari	Sos sojowy, Tamari	1.9	0.9	2.7
Zatar	Zatar	0.2	1.6	1.9
Vinegar	Ocet	0.8	0.4	1.2
Harissa	Harissa	0.2	0.8	1.0
Honey	Miód	0.9		0.9
Gyros mix	Gyros mix	0.3	0.5	0.9
Yeast flakes	Płatki drożdżowe	0.5	0.1	0.7
Tandoori	Tandoori	0.1	0.4	0.5
Maggi	Maggi	0.2	0.2	0.4
Fish sauce	Sos rybny	0.1	0.3	0.4
Five spice powder (Chinese)	Przyprawa pięciu smaków	0.1	0.3	0.4
Togaroshi	Togaroshi	0.1	0.2	0.3
Kmeli-suneli	Chmeli-suneli	0.2	0.1	0.3
Tzatziki	Tzatziki	0.1	0.2	0.3
Black pepper with powdered lemons	Pieprz cytrynowy		0.3	0.3
Herbs of Father Mathew	Zioła ojca Mateusza	0.2		0.2

**Table 3 plants-12-00591-t003:** Types of seasoning that the respondents started using less than 10 years ago. Scientific botanical names can be found in Table 1 and are provided here only for plants that were not previously mentioned.

	*N* = 916
Turmeric	22.60
Cumin	17.14
Capiscum products, including:	13.86
Smoked paprika	6.00
Gochugaru	2.84
Chili	2.40
Cayenne pepper	0.98
Chipotle	0.22
Coriander, including:	10.70
leaves	1.20
seeds	0.66
Nigella	10.37
Ginger	10.15
Green cardamom	9.28
Curry mixes	6.55
Fenugreek	5.79
Ramsons	5.68
Garam masala	5.13
Savory	4.15
Kaffir leaves	4.04
Anise	3.93
Sumac	3.93
Lemongrass	3.93
Saffron	3.71
Ground ivy	3.60
Rosemary	2.95
Nutmeg	2.62
Lovage	2.62
Thyme	2.40
Wild savory	2.29
Asafoetida	2.07
Oregano	2.07
Zatar mix	1.97
Basil	1.86
Galangal	1.86
Cinnamon	1.75
Sichuan pepper	1.75
Sage	1.64
Tarragon	1.42
Caraway	1.31
Soy sauce	1.31
Juniper	1.09
Harissa mix	0.98
Fennel	0.98
Provence herb mix	0.98
Cloves	0.87
Marjoram	0.76
Black pepper with lemon powder	0.76
Ground elder	0.76
Garlic, including:	0.76
Granulated garlic	0.22
Anise	0.66
Mustard seed	0.66
Pieprz ziołowy (“Herbal pepper mix”)	0.66
Tomatoes (dried)	0.66
Ajwain	0.55
Pellitory	0.55
Mugwort *Artemisia vulgaris* L.	0.55
Hyssop	0.55
Curry leaves	0.55
Mint	0.55
Long pepper	0.55
Sesame	0.55
Black mustard	0.44
Lavender	0.44
Bay leaf	0.44
Color pepper mixes	0.44
Vanilla pods	0.44
Black pepper, including	0.44
White pepper	0.22
Curry plant (*Helichrysum italicum* (Roth) G. Don fil.)	0.33
Himalayan salt	0.33
Khmeli suneli mix	0.33
Nettle	0.33
Parsely	0.33
Perilla	0.33
Ras el hanout mix	0.33
Svan salt mix	0.33
Tagetes spp.	0.33
Tonka beans	0.33
Wormwood (*Artemisia absinthium* L.)	0.33
Yeast flakes	0.33
Archangel	0.22
Avocado seed	0.22
Balsamic vinegar	0.22
Black cardamom	0.22
Black salt	0.22
Carrot (seeds)	0.22
Celery seeds	0.22
Dandelion root	0.22
Dill	0.22
Dittany	0.22
Fish sauce	0.22
Garlic mustard (*Alliaria petiolata* (M.Bieb.) Cavara & Grande)	0.22
Helichrysym sp.	0.22
Horseradish	0.22
Lemon verbena	0.22
Liquorice (*Glycyrrhiza glabra* L.)	0.22
Mace	0.22
Milk Thistle	0.22
Mushrooms (dried)	0.22
Nettle seeds	0.22
Onion (dried)	0.22
Rose petals	0.22
Seaweed (Algae)	0.22
Smoked salt	0.22
Solidago spp. (flowers)	0.22
Tamarind	0.22
Tansy *Tanacetum vulgare*	0.22
Wasabi	0.22
Yarrow	0.22

**Table 4 plants-12-00591-t004:** Species and types of seasoning which respondents used 10 years ago and do not use anymore.

	*N* = 916
No change clearly stated	25.76
Soup seasoning	11.90
including	
Vegeta™ brand	5.79
stock cubes	2.07
Kucharek™ brand	1.09
Maggi spice	1.86
Capsicum products	1.86
Salt	1.75
Caraway	1.64
Black pepper	1.64
including White pepper	0.66
Turmeric	1.53
Marjoram	1.09
Pieprz ziołowy (“Herbal pepper mix”)	0.98
Curry mix	0.87
Thyme	0.66
Provence herb mix	0.66
Sugar	0.55
Lovage	0.55
Garlic (dried)	0.55
Basil	0.44
Savory	0.44
Cumin	0.44
Bay leaf	0.44
Allspice	0.44
Asafoetida	0.33
Tarragon	0.33
Nutmeg	0.33
Coriander	0.33
Dill	0.33
Oregano	0.33
Spice mix for meat	0.33
Saffron	0.33
Five Spice Powder	0.22
Nigella	0.22
Wild savory	0.22
Green cardamom	0.22
Fennel	0.22
Mint	0.22
Spirit vinegar	0.22
Parsley	0.22
Spice mix for fish	0.22
Rosemary	0.22
Soy sauce	0.22
Salt with iodine	0.22

**Table 5 plants-12-00591-t005:** Chi-Squared (Χ^2^) table comparing the frequency of mention of spices in the questions about the use of spices. The expected frequency is based on the frequency of spices used at least once a year.

**What Spices/Condiments Have You Stopped Using in the Last 10 Years?** ***N* = 359 (Number of Use Reports in This Question)**	**Observed** **Mentioned**	**Not Mentioned**	**Expected** **Mentioned**	**Not Mentioned**	** *p* **
Soup seasoning	109	250	0	359	$$$c
Maggi	17	342	0	359	$$$c
Herbal pepper	9	350	1	358	$$$c
Provence herbs	6	353	3	356	NSc
Lovage	5	354	6	353	NS
Curry	8	351	7	352	NS
Caraway	15	344	8	351	$
Thyme	6	353	12	347	NS
Marjoram	10	349	14	345	NS
Turmeric	14	345	16	343	NS
Black pepper	8	351	18	341	*
Capsicums	8	351	18	341	*
**What Spices/Condiments Have You Started Using within the Last 10 Years?** ***N* = 1989 (Number of Use Reports in This Question)**	**Observed:** **Mentioned**	**Not Mentioned**	**Expected:** **Mentioned**	**Not Mentioned**	** *p* **
Asafoetida	19	1970	2	1987	***c
Soy sauce	12	1977	1	1988	***c
Zatar	18	1971	2	1987	***c
Garam masala	39	1950	7	1982	***
Harissa	9	1980	1	1988	***c
Ground ivy	33	1956	6	1983	***
Kaffir	37	1952	7	1982	***
Galangal	17	1972	3	1986	***c
Sichuan pepper	16	1973	3	1986	***c
Lemon grass	36	1953	9	1980	***
Sumac	36	1953	10	1979	***
Nigella	95	1894	28	1961	***
Ramsons	52	1937	17	1972	***
Curry	60	1929	20	1969	***
Mountain Savory	21	1968	7	1982	***
Cumin	157	1832	57	1932	***
Fenugreek	53	1936	19	1970	***
Saffron	34	1955	13	1976	***
Turmeric	207	1782	88	1901	***
Juniper	10	1979	5	1984	*
Coriander	81	1908	44	1945	***
Cardamom	85	1904	50	1939	***
Garden Savory	38	1951	28	1961	NS
Ginger	93	1896	75	1914	*
Anise	36	1953	32	1957	NS
Sage	15	1974	15	1974	NS
Tarragon	13	1976	18	1971	NS
Lovage	24	1965	37	1952	$
Rosemary	27	1962	57	1932	$$$
Nutmeg	24	1965	52	1937	$$$
Thyme	22	1967	71	1918	$$$
Mustard seed	6	1983	23	1966	$$$
Caraway	12	1977	46	1943	$$$
Basil	17	1972	74	1915	$$$
Mint	5	1984	25	1964	$$$
Oregano	19	1970	85	1904	$$$
Capsicums	22	1967	104	1885	$$$
Cinnamon	16	1973	83	1906	$$$
Cloves	8	1981	60	1929	$$$
Marjoram	7	1982	82	1907	$$$
Garlic	5	1984	63	1926	$$$
Nettle	0	1989	10	1979	$$c
Vanilla	0	1989	13	1976	$$$c
Dill	0	1989	23	1966	$$$c
Allspice	0	1989	63	1926	$$$c
Black pepper	0	1989	99	1890	$$$c

* symbol represents a significant difference showing an increase in the use of the spice, $ symbol represents a significant difference showing a potential decrease in the use of the spice. Significance levels: * or $—*p* < 0.05, $$—*p* < 0.1; *** or $$$—*p* < 0.001; ns—not significant, c—approximate significance due to numbers of less than 5 in one of the cells of the Chi-squared table.

**Table 6 plants-12-00591-t006:** A brief history of the use of spices and herbs in Poland (we included the species mentioned by at least 10% of the informants). For the scientific names of the plants see Table 1.

English Name	Brief History of Use in Poland	Century or Period When First Used as Seasoning in Poland
Capsicum (chili, paprika, etc.)	Commonly used only from the mid-19th century, mainly in the 19th c. province of Galicia, to make dishes inspired by the Hungarian goulash [62,74].	19th c.
Black pepper	Known and used since Medieval times, even mentioned in the court accounts of King Władysław Jagiełło and Queen Jadwiga from the turn of the 14th and 15th c. [53] as well as in the 14th and 15th c. accounts of the Teutonic order in Gdańsk [56]. The most universally used exotic spice in Polish cuisine. Gradually its price went down, and the spice could be used by lower strata. In contrast to other Asian spices used in Old Polish cuisine, which survived mainly as dessert seasonings, its use is restricted to savory dishes.	Medieval
Turmeric	Used since the Renaissance but mainly as a medicinal plant. First mentioned by Siennik in 1568 [47]. Its oldest names are variants of “Indian saffron”. Later only used rarely as butter and margarine coloring.	16th c. (very seldom)
Oregano	Occurs in Medieval texts but broadly used since the Renaissance as a medicinal plant and also in herbal mixes (from the 16th c.). An important folk medicinal plant in the Carpathians, but never a seasoning [15]. Oregano became popular as a seasoning only with the fashion for pizza in post-Socialist Poland.	Medieval, only medicinal, popular as a spice since the 1990s
Cinnamon	A popular exotic spice, one of the most important and most used after pepper, important both in the Middle Ages and modern cuisine. Mentioned in the 15th c. accounts of the Teutonic order in Gdańsk [57]. Often used in “Kuchmistrzostwo” (c. 1540) [40], which lists cinnamon as belonging to the basic set of spices (saffron, ginger, pepper, cinnamon, cloves) added together to meat and fish, and included in the accounts of Jadwiga and Jagiełło [53]. Used extensively for medicinal and culinary purposes, as a universal spice for meat and fish, and in flavored vodkas, especially cinnamon vodka itself. Like most other Eastern spices (except pepper), in the 18th c. its use clearly decreased and was limited to sweets, milk, flour dishes, cakes, and fruit desserts.	Medieval
Marjoram	Medicinal and culinary herbs have been frequently reported in most Polish culinary texts since the Middle Ages. Czerniecki mentions it in 1682 [37] in his list of herbs required in the kitchen, used primarily for venison, poultry, and delicate meat pâtés. In “Moda bardzo dobra” of 1686 [39], it is mentioned as a spice for pâtes and veal tripe. In 1613 Syrenius [44] described and suggested the use of marjoram confection for pearl barley soup. In the 18th and 19th c., it was used mainly for ham, sausage, cured meat, and other meat dishes.	Medieval
Ginger	One of the most popular Asian spices, considered a healing substance and a universal spice, frequent since the Middle Ages, often listed in “Kuchmistrzostwo” (c. 1540) [40] and “Compendium ferculorum” (1682) [37]; the basic ingredient of a set of hot spices popular in elite cuisine, including saffron, cinnamon, etc. It went out of use in the 18th century, and was then more often used for vodkas, desserts, or fruit dishes. Ginger is also mentioned by Paul Tremo (c. 1790), but rarely and in small numbers [20]. Mentioned in the 15th c. in the accounts of the Teutonic order in Gdańsk [56].	Medieval
Basil	Occurs in written sources since Medieval times, e.g., in “Kuchmistrzostwo” (c. 1540) lists vodka with basil flowers [40] but became more popular for meats and sausages towards the end of the 17th c., and especially in the 18th c. with the popularization of the new French culinary style, which pushed away savory spices [20,38]. Also popular in the 19th c. [59,79].	Medieval, mainly since 17th c.
Thyme	Known since Medieval times as “szmer”, but used more frequently from the 17th c., often as part of *bouquet garni* introduced by the new French cuisine [38]. Common in medicinal sources throughout.	Medieval, mainly since 17th c.
Bay leaf	Originally the fruit was used medicinally, and the leaves have appeared as a culinary ingredient since the 17th c., and especially in the 18th c. for meats, particularly meat and fish marinades, salted meat, etc. [20,39]. Present in most cookbooks from the 19th and 20th c.	17th c.
Garlic	Very widely used since the first written records appear, especially frequently during periods of fast.	Medieval
Allspice	Described as “English root” and “English pepper”, allspice was especially popular in Poland at the end of the 18th c., often in recipes describing preservation: salting fish, marinating fish, sausages, brawns, smoked goose, salted and smoked tongues, vinegar, etc., more rarely with cheese, vodka, etc.; valued for combining the taste of various spices. Allspice was considered to taste like a mixture of cloves, cinnamon, and pepper, and spread late, in association with the fashion of England at the end of the 18th c. Also recorded in 18th c. Gdańsk latrines [55].	18th c.
Cloves	One of the most popular Asian spices, a universal spice common since the Middle Ages, often in “Kuchmistrzostwo” (c. 1540) [40] and “Compendium ferculorum” (1682) [37]. Mentioned in the 15th c. accounts of the Teutonic order in Gdańsk [56]. After the use of most of the other hot spices was discontinued, cloves remained part of savory recipes for some time.	Medieval
Rosemary	Rosemary is widely and universally used as an addition to meats, fish, marinades, cured and salted meats, but also sweets, whipped cream, confections, and vodkas. A separate rosemary vodka (alcohol) has been widely used since the 16th c. Rosemary has been very popular since the 17th c. as part of *bouquet garni* and the French custom of basing flavor on herbs, not exotic spices. Czerniecki (1682) [37] considers rosemary an important addition to broth, and a rosemary soup figures in “Moda bardzo dobra” (1686) [39]. In the 19th c., it was especially often used with fish, meat, particularly game, as well as in marinades and vinegars, but it disappeared from sweet recipes.	16th c.
Cumin	Difficult to distinguish from caraway in older texts; rarely occurs in recipes. First mentioned by Syrenius [51] as “kmin kramny” (“market cumin”), used with lovage root, cabbage, and pumpkin, as opposed to “kmin polny” (caraway, literally “field cumin”). Also clearly mentioned in “Zbiór dla kuchmistrza”, in a recipe for the goose, no 321, 17th c. [40], and to flavor vodka (texts from 17th c. and 18th c.) [40]. In the 18th c. mentioned by Tremo as “kmin wenecki” (“Venetian cumin”) [20].	17th c.
Nutmeg	One of the most popular spices since the Middle Ages, basic in Renaissance and Baroque cuisine [37,38,40] mentioned by most sources; often used with a whole set of other exotic spices. Being milder in taste, it remained in use a bit longer, e.g., for dairy. Mentioned in the 15th c. accounts of the Teutonic order in Gdańsk [57]. One of the few exotic spices still used in the 18th–19th c. [20,59].	
Green Cardamom	Popular first as an ingredient in medicinal potions. One of the most important Asian spices, although used separately from the most common, classic set. Frequently used for sweets, for example in “Moda bardzo dobra” (1686) [39], for medicinal vodkas, as an important gingerbread spice, and very often as an addition to vodkas in the 18th c.; in a cookbook from 1757 [40], it is listed as a spice for mead. Used less frequently for salami, vodkas, cakes, and cookies, but still quite systematically for gingerbread.	17th c. or before
Caraway	A native plant, widely and comprehensively used. Already in the 16th c. Falimirz [46] wrote that it grew “abundantly” in Poland; at the level of sources, it is often difficult to distinguish from cumin, as the word “kmin” was used for both. The terms “Polish cumin” or “field cumin” signify caraway. It was often found in various fields of culinary art, fish, meat, cheese, pickles, biscuits, beers, spirits, liqueurs, etc. Recorded in Medieval deposits from Kraków, Poland [57]. Mentioned in the 15th c., in the accounts of the Teutonic order in Gdańsk [56]. Widely used in folk cuisine, mainly as a spice for bread, meat dishes, and sauerkraut [15].	Ancient Slavic
Coriander	Known since the Middle Ages as “Polish pepper” [43]. Mentioned in the 15th c. accounts of the Teutonic order in Gdańsk [56]. Czerniecki (1682) [37] describes it as indispensable in the kitchen, and mentions sugar flavored with coriander and its use for meats and pâtés; in the 17th c., it was also used for gingerbread. Often mentioned in “Moda bardzo dobra” (1686) [39], gingerbread was said to require coriander, and it was also used for vodka, smoked and marinated meats, and fish. In Polish cuisine, the seeds have been used for flavoring. The use of leaves is novel, influenced by Asian cuisine.	Medieval
Lovage	Commonly and widely used as a seasoning, referred to as a “familiar” (“swojski”) ingredient in recipes (1751) [80]. Syrenius (1613) [51] mentions the use of young shoots of lovage in the kitchen, like asparagus; also prepared in sugar as a confection; in the texts, the texts clearly treat lovage as a “root”, i.e., a spice with a strong, expressive taste, due to which it requires the addition of a lot of extra sugar. Used as a seasoning for mushrooms and meat. Less common in elite cuisine. In the 19th c., it almost did not appear in cookbooks, giving way to standard spices and herbs.	Pre-18th c.
Star anise and anise	Very difficult to distinguish in old texts, where it is usually called “anyż”, or “anyżek”; probably both species are present in the source texts. Additionally confused with caraway, to which these names were also sometimes applied. Used for pork jelly in “Kuchmistrzostwo” from around 1540 [40]. In the 17th c., it was usually used to flavor sugar and sweets. “Moda bardzo dobra” from 1686 [39] mentions it for frying various confections and making gingerbread. In “Kucharz doskonały” from 1783, anise is mentioned once with sausages and twice with medicinal vodkas and liqueurs [38]. In 1613, Syrenius described anise rusks [51]. Star anise was mentioned in 1793 by Tukałło [61] under the name of “badian”. In 1830, Szytller [62] advised adding star anise or cardamom to donuts. In 1841, Szyttler wrote about adding star anise to 4 thieves’ vinegar [68]; in 1845, about seasoning elk or deer brains with star anise [70]. In 1846, Szyttler advises seasoning pear pudding with star anise [71]. In the 1843 cookbook “Dwór wiejski”, it was listed as a spice for a dish called “babka chlebowa” and gingerbread. In 1856, Leśniewska [59] wrote about adding star anise to mustard, and also about “badyjanek powder” as an addition to a pig’s head or boar’s head, usually an archaic traditional Easter dish.	European anise—16th c. or before, star anise—18th c.
Savory	Quite popular in elite cuisine in the late 17th and the 18th c., savory was part of *bouquet garni*, and hence often appears in Wielądko’s “Kucharz doskonały” (1783) [38].	Pre-18th c.
Nigella	Crescentyn mentions adding nigella to bread as early as the mid-16th c. [48]. In the 16th c., white bread with nigella and anise was considered luxurious (Umiastowski, 1594) [49]. Also, Marcin from Urzędów (1595) [4] and Syrenius (1613) [51] mentioned that nigella was used as bread flavoring. In the 19th c., nigella was also used to bake bread, and especially rolls, bagels, strudels, rusks.	16th c.
Parsley	Green parsley has been recorded since the Middle Ages, e.g., in the accounts of Jadwiga and Jagiełło [53]. Czerniecki (1682) [37] uses it very often with stews, meats, fish, and soups; in “Kuchmistrzostwo” (c. 1540) [40] parsley was usually mentioned without distinction, but it was sometimes specified whether the green leaves or roots were used. This text mentions the addition of green parsley to green sauces, leafy vegetable thick soups popular since the Middle Ages, and the popularity of colorful sauces, green sauce with parsley, especially for poultry and carp in jelly. Similar coloring effects of green parsley are utilized in the “Königsberger Kochbuch” (a Teutonic cookbook from the second half of the 15th c.) [52]. Even more popular in the 18th c., during the trend for French cuisine, when it replaced sharp exotic roots; the most important ingredient of *bouquet garni*. Mentioned in “Kucharz doskonały” (1783) [38], Tremo’s book [20], in the 19th c., and generally consistently and frequently throughout the centuries.	Medieval
Mint	Medicinal plants widely used for centuries. Spiczyński (1556) [45] recommended it for mint sauce, Siennik (1568) [47] for capers and quince juice; Crescentyn (1571) [48] for use with pumpkin. In 1540 [40] for green sauces. Syrenius [51] advised adding mint to mushrooms; in the eighteenth century, it was added to milk to ensure its freshness and protect against curdling, as well as to candied calamus, vinegar, and mushrooms. In modern times, often used in various confections and liqueurs for the stomach. Often used in medicinal and herbal vodkas. In the 19th century, included in vinegars, mint liqueur, and other vodkas, oils, etc. [68]. Used to keep apples for the winter for eating or for roasting for salad, and also in Ruthenian cold soup from salted fish (Szyttler 1846) [71]. Nakwaska (1843) advises adding it to kvass or pickled cucumbers [60], Leśniewska (1856) to dumplings with cheese [59]; “Kuchnia polska” (1856) [73] to preserve groats and cheeses and, together with rue, to repel rodents. Widely used for herbal teas in SE Poland in the 20th c., and nowadays as a seasoning for “pierogi” dumplings or broth [15].	16th c. but probably much earlier
White mustard	In the past, both white mustard and *Brassica nigra* (black mustard, now obsolete in Polish cuisine) were used, cited, and often confused. Czerniecki (1682) [37] clearly mentions black mustard in his list of spices and in the recipes themselves, but sometimes it figures as simply “mustard”; similarly, both white and black mustard is noted in the handwritten “Moda bardzo dobra” (after 1686) [39]. In many recipes from the 16th–18th c., mustard was treated as a cheap, local substitute for exotic spices in simpler versions of dishes, especially with fish, various sauces or thick soups, and vegetables; a medical work from the 16th c. includes white mustard specifically in its description of the medicinal confits of eggs and wine with spices [81]; Syrenius (1613) [51] expressly mentions white mustard added to boiled turnips. Haur (1679) [50] suggests that mustard is a generic term used for various species: “This seed is of a kind of various colors, black, white and red.” In the 18th c., white mustard was used more often for vodkas or medicinal oils, and fruit was stored in white mustard seeds flooded with wine must. Szyttler wrote in 1841: “You can use flour for mustard from both white and black mustard, which we call Arabian” [68]. In the 19th c., white mustard was often referred to simply as “mustard”, and used for pickled cucumbers, mustard, sauces, etc.	Pre-16th c.
Dill	Mentioned as an important spice by Czerniecki (1682) [37] and in “Kucharz doskonały” (1783) [38]. Green dill was often mentioned in 18th c. sources. Rej (1568) [82] mentions “koprzyk” in sauerkraut. Syrenius [51] lists the use of its leaves and fruits for lactofermented cucumbers and sauerkraut as well as cured meats and vodkas.	17th c. but probably much earlier
Fenugreek	The use of the Latin name “foenum Graecum” in Old Polish times suggests its low prevalence; poorly documented, it appears in recipes for medicinal vodkas as foenum Graecum [40]. Haur (1679) [50] mentions its use for stopping wine fermentation. In the 19th c. used for cheese.	17th c. but very seldom used
Tarragon	Difficult to catch in old texts due to its variable name. Mentioned as “torun” by Syrenius (1613) [51]. Wielądko 1783 [38] calls it “*estragon*, or *toruń* herb”. Often used with meat and especially poultry, popularized in Poland during the trend for French cuisine in the 17th c., but especially in the 18th c. as part of *bouquet garni*. Wielądko [38] describes it as one of the “front herbs” (“ziela przednie” or maybe “exquisite herbs”), which included basil, thyme, bay leaves, leeks, celery, and parsley. In the 18th c., it was also added to vinegar; tarragon was regularly used by Paul Tremo [20] and, probably thanks to the popularity of his recipes, spread further.	17th c. but very seldom used
Ramsons	Not present in old recipe books and not used in traditional Polish folk cuisine. Becoming a popular food ingredient and seasoning since the late 1990s [15].	New
Sage	Already popular a long time ago, e.g., “Kuchmistrzostwo” (c. 1540) [40] mentions an interesting sage “gąszcz”, a chutney-like dish of sage leaves with carrot purée in tempura—fried. In the 16th–18th c., often used for confectionery and sweets, but also cheese, scrambled eggs, tea, sauces, vinegar, whey, vodka, and health broths. In the 19th c., it was used with meat and fish, pickled fish, smoked goose, pork hams, etc. It has also been frequent in mixed medication formulas since the Renaissance. Sage used to be commonly grown in rural gardens mainly for medicinal tea.	16th c.
Saffron	The most expensive, most luxurious spice, used since the Middle Ages, also as a food colorant for the most refined dishes. An emblem of Polish cuisine in the 17th and 18th c.; dishes containing saffron were described as “Polish”. Saffron appeared in Asian spice sets in the Renaissance and Baroque, but slightly less often due to its price [37,39,40,41,42]. Also mentioned in the accounts of the Teutonic order in Gdańsk [56]. Used throughout the centuries in fish recipes. Disappeared in the second half of the 18th c., apart from a relic recipe of saffron *baba*—a ritual Easter cake.	Medieval
Vanilla	A product from America, which spread much later and to a much lesser extent than other spices. Vanilla became a widespread addition to chocolate, but also to flour dishes, desserts, and cakes, often clearly replacing the old Asian spices, e.g., cinnamon and nutmeg, only from the mid to late 18th c.	18th c.
Nettle	Used as a leafy vegetable and as an addition to salads and soups, it seems difficult to define as a spice in the sources, but it occurs as a small addition to health broths at the end of the 18th c., as a substance for stuffing wild birds to maintain their shape and texture; in the 16th c. as an ingredient and substitute for spinach, an addition to borscht dishes in Syrenius (1613) [51], then popularized by the literature of sentimentalism and romanticism, with the fashion for folklore, wildness, and localness in the second half of the 18th c.; nettle often appears in Tremo’s records [20], and appears occasionally in those of his disciple Szyttler, but eventually goes out of official culinary use. Widely used as poor peasant food, mainly for soups and potherb, in the 19th c. and early 20th c. [15].	Ancient Slavic vegetable
Sumac	Missing in historical documentation, probably referred to in Siennik (1568) [47] as “tanner’s grain”; Kluk [58] also only mentions its use for dyeing. Listed by a few 19th c. sources, but not as food used in Poland.	New as seasoning
Lemongrass	Not recorded in historical culinary sources.	New
Kaffir lime	Not recorded in historical culinary sources.	New
Wild savory	Not recorded in traditional recipes, wild savory became known in the Socialist times during mass holidays to Bulgaria since the 1970s, where it has been widely used as one of the main seasoning ingredients.	New
Ground ivy	Listed in 1793 as “bluszcz ziemny” in a herbal tea mix and as “bluszcz poziemny” in a healthy broth recipe [61]; in 1716 as “konradek” in medicinal confectionery [83]; widely used in the folk cuisine of the 19th and 20th c. for seasoning soups, mainly in the Carpathians [15].	18th c.
Juniper	Versatile and frequently used since the Middle Ages, especially for meat, game, and wine; canned, salted, and cured meats, but also pickles. In “Kuchmistrzostwo” (c. 1540) for roast beef [40]. Tremo [20] added juniper to sawdust when smoking fish. Also, for vodkas, beers, vinegar, smoked hams, and marinades. In the 19th c., the use was narrowed down to meat, mainly venison, as well as to vodka, sauerkraut, or smoked fish. Widely used in the folk cuisine of the 19th and 20th c. In northern Poland, juniper was also fermented into a beer-like drink [15].	Ancient Slavic
Onion	Widely used in ancient cuisine since times immemorial. A basic vegetable, strongly associated with Polish cuisine since the 15th c., often cooked; the base of many sauces and purées; foreigners have remarked that Poles and Hungarians overuse onions and apply them in large quantities to everything.	Pre-15th c.
Lemon balm	Lemon balm was infrequently used as a spice; at the end of the 18th c. it was a component of medicinal broths. Mainly used as an additive to wine, thought to preserve it; occasionally used as an ingredient of vodkas and confections, and more rarely in herbal soup.	18th c.
Horseradish	Used early and widely; considered a local cheap replacement for luxurious hot spices; used mainly for meat and fish, sauces, mustards, and potherb. Horseradish leaves were also used for dyeing cheese green. It continues to be one of the most important spices [37,39,46]. Widely used in folk cuisine, especially for Easter meals [15].	Ancient Slavic
Chives	Difficult to distinguish at the level of texts, Czerniecki (1682) [37] mentioned “a young green onion”; Wielądko (1783) [38] wrote about chives during the fashion for adapting herbs from French cuisine (*bouquet garni*), when they became a universal addition with other, and local, herbs for soups, stewed meats, sauces, etc. At the end of the 18th c. tips on how to dry and preserve chives and other herbs started appearing. Chives were then added to sauerkraut. In the 19th c., they were frequently used for meats, sauces, eggs, offal, cold soups, and soups, as well as to make chive sauce. A very popular addition, but only from the second half of the 18th c.	17th c.
Galangal	First recorded in the 15th c., used in the 16th–18th c. mainly as an addition to drinks, especially the so-called hippocras—aromatic, herbal, and spiced healing wines, vodkas, medicines, and meads (Haur (1679) and then guides from the 18th c. [50]). Galangal was still present in Poland for a long time during the Renaissance and Baroque periods as an addition to vodkas and wines. In the first half of the 19th c., it was very occasionally added to vodkas and the most archaic dishes (bear paws, Szyttler [70]).	15th c. but later discontinued
Sichuan pepper	Not recorded in traditional recipes, known and available in very few Asian shops only since around 2010.	New
Sesame	Not recorded as a seasoning, rarely mentioned as a potential oil ingredient. In popular culture, known only as sweet sesame snacks available since Socialist times.	New
Asafoetida	Not recorded in Polish culinary sources, only mentioned medicinally by Syrenius (1613) as “zapaliczka” [51].	New
Lavender	Mentioned since the 16th c. in relation to the production of medicinal substances, since the 17th c. [39,40,51] in sweet confections, syrups, vinegars, and as an addition to beer, wine, vodkas, and gingerbread. There is also a single example of its use in herbal soup. Sometimes the lavender flower is mentioned. Used in the 19th c., though rarely, e.g., for Toruń gingerbread [68].	17th c.
Celery	Widely used as a root vegetable rather than a condiment; an ingredient of soups and combined with meat and fish, as in Czerniecki 1682 [30], sometimes used as “zaprawa”—a flavor additive in the form of purée or sauce, as listed in 1783 [38]. There is also a rare record from 1821 of the use of celery seeds as a spice for vodka [84]. Celery continued to be used primarily as a root vegetable throughout the 19th c., also mainly as a root vegetable, but the leaves were also used, among others for pickling cucumbers. Disseminated later than other root vegetables, widespread in Poland probably only from the 17th c.	17th c. but mainly as vegetable
Mushrooms (dried)	Common and frequent use in Polish cuisine. Czerniecki (1682) [30] advises using mushroom broth as a seasoning for meat stews, and dried mushrooms for the best French pottage. Czerniecki’s [37] “seventh condiment” was a sauce with a flavor additive made of dried mushrooms and green parsley. This additive was described later in the 18th c., often used with fish, and an important ingredient of fish soups and fasting dishes in general. Wielędko [38] describes a spice made from dried mushrooms. In the 18th c., it was also used for pâtés, meats, stews, etc. Also widely used in folk cuisine of the 19–21st c., and probably present since ancient times, as it is part of the most traditional Christmas Eve dishes in the whole country.	Ancient Slavic
Ground elder	Mentioned as a leafy vegetable, analogous to sorrel or kale (in 1706 as “śnitka”), often mentioned (as “giers”) at the court of Sigismund III in 1631; used analogously to sorrel, green parsley, and spinach [54].	Pre-17th c., probably an ancient Slavic vegetable but new as a seasoning
Hyssop	References to hyssop appear in the Middle Ages in reference to biblical texts; from the 16th c. it was included in medicines and healing potions; in the 17th and 18th c. it was added as a spice, e.g., for rice cakes [51], goat whey [42] and vodka flavoring [40]; since the 19th c. extremely rarely used as a flavoring for vinegar.	17th c.
Fennel	Fennel is often difficult to distinguish from dill in texts, as both are called “koper” in Polish. The seeds, roots, herbs, and whole pickled shoots have been used in cuisine. Very popular in the 16th–18th c., then less so. Siennik (1568) [47] uses it with chicken broth, Syrenius (1613) [51] writes about fennel root included in soup; Falimirz (1534) [48] includes it in his general rules for preparing fish, advising the addition of “Italian fennel or Polish fennel” (probably seeds); he also recommends adding it to salads (herbs) and as an ingredient of “spinach”; similarly, Siennik (1568) [47] proposes it as an ingredient of “salsza” sauce for greens; Rej [82] for beetroots, often preserved in vinegar; Syrenius (1613) [51] writes about pickling whole shoots, and adding them to pickled asparagus or lacto-fermenting cucumbers. In the 16th–17th c., fennel was used for confectionery, vodkas, and beverages more generally, as well as for mustards; in 1716, fennel root was used for goat’s whey and to flavor wine [83]; in 1660 its fruits were added to prunes, cakes, and rusks [42]. In the 19th c., fennel was added to soups, fish, and mustard, but clearly less frequently than before.	16th c.
Tomatoes (dried and powdered)	Not recorded in old culinary books as a spice. In general, tomatoes are a very late import in Polish cuisine. They became a common food only in the second half of the 20th c. Tomato soup is first mentioned in 1830 (Szyttler [62]).	Used in recipes since the 19th c. but new as a seasoning

## Data Availability

The original data matrix with the answers was deposited in the Repository of Rzeszów University [118].

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
