# Peer review of "Turmeric and Cumin Instead of Stock Cubes: An Internet Survey of Spices and Culinary Herbs Used in Poland Compared with Historical Cookbooks and Herbals"

_plants, 2023, doi:10.3390/plants12030591_

Round 1
Reviewer 1 Report
Manuscript worthy of publication in Plants after major revision - all comments attached. Please provide a certificate of professional proofreading of the text.

Author Response
Please provide a certificate of professional proofreading of the text.
>The text was proofread by Nasim Łuczaj, the relative of the first author who is a British native speaker born in England, professional proofreader, and graduated from Literature studies in Glasgow University.
please write "th" in superscript - use throughout the manuscript
>we corrected the text
worth explaining Maggi
>we added an explanation
removing spacing between paragraphs - too much fragmentation of the text
>this is the MDPI template
It is necessary to emphasize the novelty of this article.
> We added: It must be emphasized that we have not found any study looking at spice use more diachronically: comparing the contemporary use of spices in any country with the his-torical data on spice use from the same area.
Please put forward a research hypothesis.
>We added a hypothesis: Our hypothesis was that some of the spices which are contemporarily used are completely new to Polish cuisine and there are no traces of their use in older cookbooks, ethnographic reports or other culinary sources.
Please accumulate the information contained in the table, e.g. by reducing the size. It's hard to read right now.
>I am sure the editing team will make the tables more compact e.g. by presenting tables 3 and 4 in two columns which will lessen the number of pages
Please add pictures (please adjust the same size of the photos) of all spices by adding one more column. It may be worth changing the orientation to landscape then.
> this is generally not practiced in scientific publications.
Reviewer 2 Report
The authors address a topic of considerable interest which is condiments or spices. In general terms, the introduction of the use of condiments is quite good, however global and later local statistics would be missing, as the results only refer to the population of Poland. The results are interesting from the local point of view, however, I consider that there are no statistics applied to it, only the results obtained are described. The discussion is quite informative, but you should consider projections of local or regional consumption relative to world demand. The results can be analyzed in greater depth, in terms of impact, however, the document does not present it. The methodology section does not indicate the type of statistical analysis used, nor does it have the questionnaire that was applied or the method applied or scale applied.
Author Response
The authors address a topic of considerable interest which is condiments or spices.In general terms, the introduction of the use of condiments is quite good, however global and later local statistics would be missing, as the results only refer to the population of Poland.
>actually these statistics are difficult to follow as many statistical sources do not specify single species. In Poland the statistical yearbooks of import do not even distinguish spices from coffee and tea. We found however a very valuable reference from India which is the largest exporter of spices in the world and we included it in the discussion showing that Indias exports of spices tripled over 2001-2017 signifying growing global interest.
The results are interesting from the local point of view, however, I consider that there are no statistics applied to it, only the results obtained are described.
> the main focus of study was historical diachronic descriptive analysis but now we explained the statistics in Methods, added an extra Chi-squared test analysis
The discussion is quite informative, but you should consider projections of local or regional consumption relative to world demand.
>these figure are difficult to obtain but we found however a very valuable reference from India which is the largest exporter of spices in the world and we included it in the discussion showing that India’s exports of spices tripled over 2001-2017 signifying growing global interest.
The results can be analyzed in greater depth, in terms of impact, however, the document does not present it.
>we agree, the analysis could be extended on many levels but we focus on the historical aspects. We may consider analyzing other aspects in another paper. However, we introduced Chi-square testing (Table 5) and an “index of change” illustrating the temporal change of spice use according to the respondents (Figure 1).
The methodology section does not indicate the type of statistical analysis used, nor does it have the questionnaire that was applied or the method applied or scale applied.
>now we explained the statistics in Methods, added an extra Chi-squared test analysis and attached the original questionnaire in additional data.
Round 2
Reviewer 1 Report
Accept in present form.
Author Response
thank you! We now made just a few small typo-type changes, e.g. anise to anise.
Reviewer 2 Report
The manuscript improved considerably, however, it still has some grammatical errors and omissions (table 65?).
Author Response
thank you! We now made just a few small typo-type changes, e.g. anise to anise and we corrected the problem with Table 5.